# Why People Do Not Attend Health Screenings: Factors That Influence Willingness to Participate in Health Screenings for Chronic Diseases

**DOI:** 10.3390/ijerph17103495

**Published:** 2020-05-17

**Authors:** Shih-Ying Chien, Ming-Chuen Chuang, I-Ping Chen

**Affiliations:** 1Institute of Applied Arts, National Chiao Tung University, Hsinchu 30010, Taiwan; cming@faculty.nctu.edu.tw (M.-C.C.); iping@faculty.nctu.edu.tw (I.-P.C.); 2Institute of East Asian Studies, University of California, Berkeley, CA 94704, USA

**Keywords:** chronic diseases, community-based health screening, participation rate, sociodemographic, willingness to continue to participate

## Abstract

*Background:* Chronic diseases are a leading cause of morbidity and mortality worldwide, and preventative screenings are the most effective way to reduce the risk of developing a chronic disease. However, many individuals do not take advantage of preventative screening services for chronic diseases, especially in rural areas. In this study, we investigated (1) the factors that affect people’s willingness to participate in chronic disease screenings and (2) reasons why people have not undergone screening for a chronic disease in the past. *Methods:* Study participants (aged 30–65 of years age; *n* = 204) included individuals from four areas in northern of Taiwan that are considered to have a high chronic disease risk. To identify factors that influence willingness to attend health screenings, data were collected by questionnaire. *Results:* Over 50% of participants (58.33%; *n* = 119) indicated that they were unaware of community-based screenings for chronic diseases offered by Chang Gung Memorial Hospital, which is one of the top-rated medical centers in Taiwan. Factors that increase willingness to participate in health screenings for chronic diseases include: (1) the convenience of screening site locations; (2) affordability; and (3) other considerations related to healthcare providers and diagnostic facilities (e.g., reputation, degree of modernization, etc.). Conversely, factors that reduce willingness to participate in health screenings include: (1) a belief that one was currently healthy; (2) lack of time; (3) a belief that screening procedures were too complicated to understand; (4) physical pain or negative emotions such as fear, anxiety, embarrassment, pain, and discomfort and, (5) having had a negative experience during a previous health checkup. *Conclusions:* Our findings demonstrate that health attitudes, sociodemographic factors, and other motivating and preventative factors affect peoples’ willingness to participate in health screenings. The motivating factors and barriers for people to participate in health screening for chronic diseases are very heterogeneous. However, understanding the barriers and motivating factors to health screening would mean that interventions with the purpose of decreasing people’s health risks and reducing deaths and disabilities caused by a chronic illness could be implemented.

## 1. Introduction 

By 2020, more than 1 billion people around the world have reached old age [1,2,3]. As a result of these changing demographics, since 2020, more than half of global healthcare expenditure has been related to the treatment of chronic conditions [1,4,5,6,7,8,9,10]. For example, approximately 45% of all Americans (133 million) suffer from at least one chronic disease [11,12,13,14,15], and one in three adults has two or more chronic conditions [16,17,18,19,20]. It is estimated that, by 2050, 21 million people in the United States will be living with a chronic disease [21,22]. Furthermore, chronic diseases such as cancer, heart disease, and diabetes, which are already leading causes of deaths worldwide [23,24,25], are becoming more widespread [11,12,26]. For example, someone is diagnosed with heart disease approximately every 40 s [27,28,29]. Moreover, nearly 8.5% of the global population is affected by diabetes, and one diabetes patient dies every six seconds as a result of the disease [30]. As in other countries, in Taiwan, an aging population has presented many challenges that have placed a strain on healthcare resources.

Routine health screening is considered to be one of the keys to reducing healthcare burdens associated with chronic diseases [31,32,33]. Health screenings can prevent and detect diseases in earlier, more treatable stages. After screening, appropriate preventive treatment is necessary. This would significantly reduce the risks posed by diseases, including disability and early death [34], and also reduce the cost of medical care. In Taiwan, several organizations provide similar health screening services. However, many people do not participate in health screenings; thus, many chronic diseases are not diagnosed before symptoms appear [35,36,37]. This is especially true in rural areas [38,39,40].

Understanding the factors that both motivate and prevent people from participating in health screenings is essential. Furthermore, a better understanding of the factors that motivate people to utilize screening services should improve satisfaction rates and promote compliance with follow-up treatment protocols [41]. Accordingly, the objective of this study is to identify the factors that have motivated and prevented individuals from participating in health screening services for chronic diseases over five consecutive years. We also sought to identify effective ways of increasing participation in preventative health screenings at Chang Gung Memorial Hospital. In so doing, we focused on sociodemographic factors (selected areas where people are suspected with high risk of metabolic diseases, kidney and lung diseases and cancers), factors related to health attitudes and health awareness, health service design, and patients’ general knowledge about chronic diseases.

## 2. Methods 

### 2.1. Study Design, Sample, and Recruitment

We adopted a community-based cross-sectional study design. A total of 204 (randomly sampled) people agreed to participate, and the study was conducted from May to November of 2019. Participants, who were 30–65 years of age, included individuals from one urban district and three rural districts in the north of Taiwan—specifically, the Anle, Ruifang, Gongliao, and Wanli Districts of New Taipei City. 

In identifying factors associated with non-participation in health screenings, we asked participants to complete a one-to-one interview and a questionnaire. Those who were diagnosed with chronic diseases and under routine clinical treatments were excluded from this study.

The sociodemographic and clinical characteristics of participants (by district) are shown in Table 1.

Participants were divided into rural and urban groups. Questionnaire answers from urban and rural participants were compared in order to identify group differences. The first part of the questionnaire sought to obtain basic sociodemographic information, including the gender, age, education level, and marital status of participants. The questionnaire also sought to obtain information pertaining to previous medical examination experiences and to determine whether participants had a healthy weight. Participant weights were classified according to the body mass index (BMI), in which weight ranges include underweight (under 18.5 kg/m^2^), normal weight (18.5 to 25 kg/m^2^), overweight (25 to 30 kg/m^2^), and obese (over 30 kg/m^2^). The second part of the questionnaire aimed to identify the factors that influence the decision to participate in a health screening. We were interested in (1) health consciousness, (2) factors that motivate and prevent utilization of health screening services, and (3) willingness to participate in future health screening services. 

Specifically, the second part of our questionnaire sought to determine the attitudes and opinions of participants. For this, participants were asked to indicate their level of agreement with a given statement using a five-point Likert scale (wherein one represented “strongly disagree” and five represented “strongly agree”). The scope of these questions included three domains which influence decision making pertaining to chronic disease screening. In the last part of the questionnaire, participants were asked to answer a few questions about chronic diseases so that we could determine their level of health knowledge. 

### 2.2. Statistical Analysis

In this study, raw data were analyzed using the SPSS statistical software package (SPSS Inc., Chicago, IL, USA). Sociodemographic data (collected using categorical variables) were summarized using total numbers and percentages, whereby differences among rural and urban populations were examined, and among subjects with higher education levels. Data were analyzed using descriptive statistical tests and Pearson correlation coefficients to identify variables which significantly influenced willingness to participate in health screenings and multivariable models were used to determine which variables were significantly influence the participants’ willingness to participate in health screening. The partial non-response rate (i.e., questionnaires which were missing data) in this study was less than 1%.

### 2.3. Ethics Committee Approval

This study was approved by the Research Ethics Committee for Human Subject Protection of National Chiao Tung University on 5 December 2018 (REC reference: NCTU-REC-107-072-e). Written, informed consent was obtained from each participant.

## 3. Results

### 3.1. Sociodemographic Characteristics of Participants

A total of 204 individuals aged 30 to 65 years agreed to participate in this study. Among these participants, 62.7% were female, and the majority (24.5%) were aged 41–50 years. None of the participants had ever participated in a community-based screening for any chronic disease at the Chang Gung Memorial Hospital, which is one of the top-rated medical centers in Taiwan. Furthermore, more than 50% of participants (58.33%; *n* = 119) said that they had never heard of these community-based screening services. 

The basic demographic characteristics of study participants are summarized in Table 2. A slim majority of participants (50.5%; *n* = 103) were from the Ruifang, Gongliao, and Wanli Districts of New Taipei City (all of which comprise rural areas), whereas 49.5 (*n* = 101) lived in an urban area (Anle). As shown in Table 2, over 40% of participants (41.7%; *n* = 85) were highly educated (bachelor’s degree or above). The two groups (i.e., rural and urban) were similar in terms of marital status: for the urban group, 67.3% (*n* = 138) were married, whereas for the rural group, 66% were married. 

In addition, more rural group participants had undergone a previous health examination (74.5%) than urban group participants (52.5%). Overall, nearly 50% of participants were overweight or obese (48%; *n* = 98).

### 3.2. Correlations between Health Attitudes and Health Behaviors Pertaining to Chronic Disease Prevention 

#### 3.2.1. Health Perceptions and Internal Motivations for Undergoing a Health Screening 

More than half of participants (55.4%; *n* = 113) reported that they may have health problems, and 46.1% (*n* = 94) of participants believed that they were at high risk of developing a chronic disease in the future. Nearly 40% of participants reported that not feeling well was the primary reason they would consider attending a health screening (39.3%; *n* = 80). Over one-fourth of the 204 study participants (26.9%; *n* = 55) said that they health screening services made them fearful because they were emotionally unable to confront disease threats. A large number of participants (76.5%; *n* = 156) believed that health problems would affect their social networks, career, and family life. Nonetheless, a majority of participants (93.6%; *n* = 191) believed that preventive health checks could be an effective way of determining their current health status, and 90.7% of participants (*n* = 185) agreed that screening can help doctors diagnose and treat chronic diseases early (i.e., before symptoms appear). Many participants also agreed that regular health checks were likely to reduce future healthcare costs (80.4%; *n* = 164) and that early disease diagnosis would increase the chances of successful treatment. Finally, the majority of participants (71.5%; *n* = 146) indicated a willingness to undergo health screenings in the future, and most participants (73%; *n* = 149) said that, if they were diagnosed with a chronic disease, they would choose to be treated by the same organization that had provided the health screening service.

#### 3.2.2. Factors that Motivate Individuals to Undergo Health Screening for Chronic Diseases

Most of the participants (85.3%; *n* = 174) reported that a convenient health screening site location would have an important influence over their decision to utilize health screening services. Cost was another key consideration, whereby 72% of participants (*n* = 147) said that cost would have the greatest influence over their decision to utilize health screening services. Furthermore, nearly 40% of participants believed that health screenings that individuals had to pay for themselves were likely to be more effective than free health screenings (*n* = 81). A large number of participants (81.4%; *n* = 166) also reported that issues related to human factors and diagnostic facilities (e.g., the sex, skills, attitudes, etc., of the healthcare professional who administers the screening) were important and influenced their willingness to participate in health screening services.

#### 3.2.3. Factors that Prevent Participation in Health Screening Services for Chronic Diseases

A significant number of participants (63.3%; *n* = 129) reported that they were currently quite healthy, and most participants (61.3%; *n* = 125) thought that health screenings were a waste of time. Nearly half of participants (47.5%; *n* = 97) said that procedures involved in health screenings are too complicated and difficult to understand. In addition, 32.8% of participants (*n* = 67) reported that certain health screening procedures or health screening items led to physical or emotional pain/discomfort. Similar findings have been observed in previous research [42,43,44], which reported that mammograms and cervical smears were likely to cause fear, anxiety, embarrassment, pain, and/or discomfort in patients. In addition, a very low percentage of participants (16.1%; *n* = 33) reported that they would refuse to participate in health screenings due to negative attitudes and behaviors of healthcare workers, and 20.1% of participants (*n* = 41) said that they had had a negative experience during a previous health checkup (Table 3).

#### 3.2.4. Willingness to Participate in Future Health Screenings

Most participants (71.6%; *n* = 146) reported that they were willing to participate in future health screenings for chronic diseases; only 28.0% (*n* = 57) of participants said that they would not consider attending future health screenings (Table 4).

The effects that individual factors were found to have on health behaviors are shown in Table 5. Health behaviors were positively related to health attitudes and health awareness, which included health beliefs and anxiety. For example, many elderly people who reported being frequently anxious about the state of their health said that they would be unwilling to participate in health screenings for chronic diseases, even if they display minor symptoms. There are other important motivating and preventative factors that influence willingness to participate in health screenings—for example, we found that, when participants believed they could receive tangible benefits from health screening services, they were more likely to be willing to take advantage of those services.

By using multivariable analysis (Table 6), a convenient location was identified to be the most important factor which influenced non-participation in community-based health screenings (OR = 0.514, 95% CI = 0.319–0.829, *p* = 0.006 *). 

Finally, it is also worth noting that health knowledge has important causal influences over behavior. Accordingly, previous research has shown that knowledge about chronic diseases has an important influence over health behavior [12]. In our study, it was found that, when a patient lacks knowledge about chronic diseases, they are less likely to participate in health screenings. Overall, participants had an average score of 35.93 points (a perfect score was 100 points) for their knowledge of chronic diseases. We did not observe any significant differences between rural and urban groups in terms of knowledge of chronic diseases: the average score for participants in the urban group was 33.27, and the average score for participants in the rural group was 38.54. These findings indicate that a lack of knowledge pertaining to chronic diseases reduces participation in health screening services.

## 4. Discussion 

### Factors that Affect Willingness to Participate in Chronic Disease Screenings

A few previous studies (conducted in multiple countries) have reported on sociodemographic differences and factors that could potentially affect the decision to undergo preventive health screening [45]. The results of the current study revealed that, in addition to sociodemographic characteristics and other factors, health attitudes and health awareness can also affect willingness to participate in health screenings for chronic diseases. Nonetheless, previous studies have paid little attention to (1) participants’ requirements for health screening and (2) how factors that predict individual behavior affect willingness to participate in health screenings [45]. In the current research, many respondents (71.6%) indicated that they would be willing to participate in future health screenings for chronic diseases. Considering a greater number of key factors when designing health screening services would likely (1) motivate an even greater number of people to participate in community-based health screenings and (2) increase the effectiveness of these services. Furthermore, participants who were middle-aged and had achieved a higher level of education were more likely to participate in health screenings, even if they had undergone previous health checkups. In addition, most participants in this study reported that they had never heard of community-based screenings for chronic diseases offered by Chang Gung Memorial Hospital. Lack of information was one of the main reasons that they had not undergone a screening in the past. In this study, we found that sociodemographic differences led to divergent opinions about and demands for community healthcare services.

Many motivating and preventative factors were found to affect decisions about whether to undergo preventive health screening. Therefore, the design of health screening services should be improved, and barriers which limit access to these services should be reduced. The motivating factor that commonly affected participant willingness to attend health screenings was the convenience of screening site location. This finding indicated that long distance transportation negatively influenced the decision to participate in health screenings. Another important factor cited by participants was the availability of incentives, such as opportunities to attend free health screening services or receive other free health benefits. Some participants in this study also reported that bad experiences during previous health checkups (e.g., long wait times) negatively influenced their willingness to attend a health screening, which is consistent with findings reported in a previous review by Dickinson [46]. All preventative factors are problematic because they prevent patients from having chronic diseases diagnosed early. 

Conversely, many participants reported that, at the time of questionnaire completion, they felt quite healthy (63.3%). Furthermore, over half of the participants indicated that they believed health screenings were a waste of time, money, and other healthcare resources. Additional preventative factors included a belief that health screening service procedures were too complicated, and this was especially true for elderly people (who are more likely to have difficulty following instructions). Some participants further reported that certain screening items were likely to cause negative feelings, such as distress. Unexpectedly, a few female participants reported that their husbands disapproved of health screenings, a preventative factor that was not identified other studies.

A few participants reported that health providers had had a bad attitude during past health checkups, and previous studies demonstrated that the attitudes of health providers have an important influence over willingness participate in health screenings [42,43,44,45]. Furthermore, in this study, many participants indicated that they believed they had a high potential risk of developing chronic disease in the future. Some participants further believed that a chronic disease would greatly affect their daily lives and indicated that they would attend a health screening when they started to feel ill. This finding suggests that study participants had misconceptions about chronic diseases and limited knowledge about health screening services, for example they may not know what body mass index (BMI) and high blood pressure is, which may influence their willingness to participate in health screening. Although most participants agreed that health screenings were useful in detecting diseases during the early stages, a few remained unwilling to participate in health screenings because they were afraid to discover that they had a serious medical condition. The factors influenced willingness to participate in health screenings are summarized in Table 7.

In Table 4, the majority of participants indicated that, if more tangible benefits were offered and/or if certain key considerations were improved, they would be willing to participate in future health screenings because they believed that regular health screenings were likely to reduce future healthcare costs. Finally, participants indicated that they were likely to seek treatment from the same organization that had provided the chronic disease screening service.

Our findings are subject to three limitations. Firstly, in this study, health-related behaviors were self-reported and therefore there may be a reporting bias. Secondly, those who consider that they might have health problems are more likely to participate in health screening. Thirdly, because areas have lower population densities than urban areas, we included subjects from three different rural areas. However, we did not consider demographic differences among rural areas. Furthermore, the participants in our study may not be representative of all rural populations, and not all rural populations in rural counties are the same.

## 5. Conclusions

This study investigated how sociodemographic differences (among three rural areas and one urban area) affect access to and use of health screening services for chronic disease in Taiwan. 

This study also provided evidence that health behaviors are positively related to health attitudes and health awareness, which in turn motivate and/or prevent people from utilizing health screening services. When people are aware that they are at risk of a chronic disease, they are more likely to take steps to prevent that disease. In addition, most participants believed that being diagnosed with a chronic disease would be extremely difficult for their families, partly due to the significant financial impacts associated with such diseases. Participants also believed that a chronic disease diagnosis would significantly affect their emotions and behaviors, which would in turn impact their social networks and family relationships. Finally, in our study, participants had incredibly low scores for knowledge of chronic diseases, and this could be the primary reason that people do not undergo important health screenings.

There are a great variety of factors that motivate and prevent people from participating in health screenings for chronic diseases. Accordingly, when developing health screening services, factors pertaining to the specific populations that these services target (e.g., sociodemographic characteristics, individual health behaviors, health attitudes and health awareness, and other motivating and preventative factors) should be considered [47].

It is not possible to develop a ‘one size fits all’ approach to the prevention of chronic diseases. Nonetheless, increasing participation in health screening services could improve early detection rates of chronic diseases. Understanding the factors that motivate and prevent people from undergoing health screenings should help reduce risks, disabilities, and mortality rates associated with chronic diseases.

## Figures and Tables

**Table 1 ijerph-17-03495-t001:** Clinical characteristics of urban and rural areas considered in this study (*n* = 204).

District	Sociodemographic Characteristics	Clinical Characteristics	*n*
Anle	Urban area (general population)	Various chronic diseases related to metabolic dysfunction	101
Ruifang	Rural area (miners)	High risk of lung, liver, and kidney diseases	103
Wanli	Rural area (fishermen, high level of alcohol use)	High risk of liver disease and metabolic syndrome	
Gongliao	Rural area (close to a nuclear power plant, retired fishermen)	High risk of liver disease, metabolic syndrome and various cancers	

**Table 2 ijerph-17-03495-t002:** Basic demographic characteristics of the 204 study participants.

Sociodemographic Characteristic	Rural Group (*n* = 103)		Urban Group (*n* = 101)		Overall Population (*n* = 204)	
	*n*	%	*n*	%	*n*	%
Sex						
Male	32	31	44	44	76	37.3
Female	71	69	57	56	128	62.7
Age						
≤30	18	17.5	14	13.9	32	15.7
31–40	14	13.6	23	22.8	37	18.1
41–50	30	29.1	20	19.8	50	24.5
51–60	20	19.4	17	16.8	37	18.1
≥60	21	20.4	27	26.7	48	23.5
Education						
Illiterate	4	4	2	2	6	2.9
Elementary school	9	9	12	11.9	21	10.3
Junior high school	12	11.8	13	12.9	25	12.3
Senior high school	37	36	29	28.7	66	32.4
University	34	33.3	40	39.6	74	36.3
Postgraduate	6	5.9	5	5	11	5.4
Missing data *n* = 1	-	-	-	-	1	0.4
Marital Status						
Married	68	66	68	67.3	136	66.7
Unmarried	35	34	33	33.7	68	33.3
Previous medical examination						
Yes	76	74.5	53	52.5	129	63.2
No	26	25.5	48	47.5	74	36.3
Missing data *n* = 1	-	-	-	-	1	0.4
Weight scale						
Underweight	4	4	5	5	9	4.4
Standard	53	51.4	44	43.6	97	47.6
Overweight	22	21.3	30	29.7	52	25.5
Obese	24	23.3	22	21.8	46	22.5
Aware of community-based screening services for chronic diseases offered by Chang Gung Memorial Hospital (CGMH)						
Yes	34	33	50	49.5	84	41.17
No	67	65	52	51.4	119	58.33
Missing data *n* = 1	-	-	-	-	1	0.5

**Table 3 ijerph-17-03495-t003:** Factors that influence the decision of participants to participate in a health screening.

Statement	Strongly Disagree (%)	Disagree (%)	Neutral (%)	Agree (%)	Strongly Agree (%)
Attitudes and beliefs pertaining to health status and health behaviors					
I might have health problems.	9 (4.4)	56 (27.5)	26 (12.7)	102 (50.0)	11(5.4)
I think I have a high potential risk of developing chronic disease(s).	3 (1.5)	51 (25.0)	56 (27.5)	73 (35.8)	21 (10.3)
I will attend a health screening if and when I feel physical discomfort.	8 (3.9)	93 (45.6)	23 (11.3)	75 (36.8)	5 (2.5)
I am afraid that I will learn I have a health problem after I participate in a health screening.	17 (8.3)	106 (52.0)	26 (12.7)	46 (22.5)	9 (4.4)
Health problems will have consequences for my social networks, job, and family life.	4 (2.0)	29 (14.2)	15 (7.4)	112 (54.9)	44 (21.6)
Preventive Health screenings could determine my current health status.	-	3 (1.5)	10 (4.9)	155 (76.0)	36 (17.6)
Preventive Health screenings could detect early warning signs of more serious diseases.	-	7 (3.4)	12 (5.9)	152 (74.5)	33 (16.2)
Regular check-Ups may reduce future healthcare costs.	-	12 (5.9)	28 (13.7)	132 (64.7)	32 (15.7)
I agree that early detection and early treatment are beneficial.	1 (0.5)	2 (1.0)	14 (6.9)	81 (39.7)	106 (52.0)
If I were diagnosed with a chronic disease, I would seek treatment from the same organization that provided the screening service.	-	26 (12.7)	29 (14.2)	103 (50.5)	46 (22.5)
Factors that motivate patients to participation in health screening services for chronic diseases					
A convenient location is important for me.	-	13 (6.4)	17 (8.3)	130 (63.7)	44 (21.6)
Paid health screenings are more effective than free health screenings.	7 (3.4)	46 (22.5)	70 (34.3)	61 (29.9)	20 (9.8)
I believe that health Service providers and facilities are well-equipped.	-	6 (2.9)	32 (15.7)	157 (77.0)	9 (4.4)
I had a negative experience during a previous health checkup.	8 (3.9)	92 (45.1)	63 (30.9)	33 (16.2)	8 (3.9)
Factors which prevent participation in health screening services for chronic diseases					
Statement	Very poor (%)	Poor (%)	Fair (%)	Good (%)	Excellent (%)
Self-assessed health status	1 (0.5)	40 (19.6)	34 (16.7)	126 (61.8)	3 (1.5)
Statement	Strongly disagree (%)	Disagree (%)	Neutral (%)	Agree (%)	Strongly agree (%)
I think health screenings are a waste of time.	1 (0.5)	40 (19.6)	38 (18.6)	115 (56.4)	10 (4.9)
I think that health screening procedures are complicated.	6 (2.9)	67 (32.8)	34 (16.7)	89 (43.6)	8 (3.9)
Some health screening items lead to fear, anxiety, embarrassment, pain and/or discomfort.	9 (4.4)	80 (39.2)	48 (23.5)	58 (28.4)	9 (4.4)
Some healthcare providers have negative attitudes.	13 (6.4)	109 (53.4)	49 (24.0)	27 (13.2)	6 (2.9)

**Table 4 ijerph-17-03495-t004:** Willingness to undergo future health screenings for chronic diseases.

Statement	Rural Group (*n* = 103)		Urban Group (*n* = 101)		Overall Study Population	
I am willing to participate in future health screenings						
Yes	76	73.8%	70	69.3%	146	71.6%
No	32	31.1%	25	24.8%	57	28%
Missing data *n* = 1	-	-	-	-	1	0.5%

**Table 5 ijerph-17-03495-t005:** Correlations between health attitudes, health awareness, and health behaviors pertaining to chronic disease prevention (*n* = 204).

Statement	Factors Influencing Non-Participation in Community-Based Health Screenings	
	Sig. (2-tailed)	Pearson Correlation Coefficient
Health attitudes and health awareness		
I might have health problems.	0.000	0.578 **
I think I have a high potential risk of developing health problems.	0.000	0.743 **
I do not attend health screenings because they make me feel uncomfortable.	0.000	0.683 **
I afraid I will learn that I have health problems after I participate in a health screening.	0.000	0.587 **
Health problems will have consequences for my social networks, career, and family life.	0.000	0.799 **
Preventive health screenings could determine my current health status.	0.000	0.869 **
Preventive health screenings could detect early warning signs of serious diseases.	0.000	0.777 **
Regular check-ups may reduce future healthcare costs.	0.000	0.727 **
I believe in early detection and early treatment.	0.000	0.737 **
If I were diagnosed with a chronic illness, I would seek treatment at the same organization that provided health screening services.	0.000	0.752 **
Factors that motivate individuals to participation in health screenings for chronic diseases		
A convenient location is important to me.	0.000	0.416 **
Cost greatly affects my willingness to participate in a health screening.	0.000	0.434 **
Paid health screenings are more effective than free health screenings.	0.000	0.593 **
I believe that health service providers and facilities are well equipped.	0.000	0.428 **
I had a negative experience during a previous health checkup.	0.001	0.234 **
Factors that prevent participation in health screenings for chronic diseases		
Self-assessed health status	0.000	0.897 **
I think health screenings are waste of time.	0.000	0.793 **
I think that health screening procedures are complicated.	0.000	0.755 **
Some health screening items leads to fear, anxiety, embarrassment, pain, and/or discomfort.	0.000	0.638 **
Some healthcare providers have negative attitudes.	0.000	0.549 **

*** p* < 0.01.

**Table 6 ijerph-17-03495-t006:** Factors influencing non-participation in community-based health screenings (*n* = 204).

Statement	Factors Influencing Non-Participation in Community-Based Health Screenings		
	OR	95% (CI)	*p*-Value
Health attitudes and health awareness			
I might have health problems.	0.790	0.505–1.235	0.301
I think I have a high potential risk of developing health problems.	0.992	0.638–1.542	0.971
I attend health screenings because I am not feeling well.	1.136	0.765–1.688	0.528
I afraid I will learn that I have health problems after I participate in a health screening.	1.098	0.746–1.615	0.635
Health problems will have consequences for my social networks, career, and family life.	0.917	0.822–1.350	0.662
Preventive health screenings could determine my current health status.	0.987	0.359–2.717	0.980
Preventive health screenings could detect early warning signs of serious diseases.	0.508	0.232–1.116	0.092
Regular check-ups may reduce future healthcare costs.	1.530	0.822–2.848	0.179
I believe in early detection and early treatment.	0.905	0.471–1.738	0.764
If I were diagnosed with a chronic illness, I would seek treatment at the same organization that provided health screening services	1.083	0.722–1.624	0.700
Factors that motivate individuals to participation in health screenings for chronic diseases			
A convenient location is important to me.	0.514	0.319–0.829	0.006
Cost greatly affects my willingness to participate in a health screening.	0.965	0.650–1.435	0.862
Paid health screenings are more effective than free health screenings.	1.370	0.947–1.982	0.094
I believe that health service providers and facilities are well equipped.	0.834	0.427–1.629	0.595
I had a negative experience during a previous health checkup.	0.801	0.532–1.206	0.288
Factors that prevent participation in health screenings for chronic diseases			
Self-assessed health status	1.116	0.659–1.891	0.683
I think health screenings are waste of time.	1.297	0.814–2.066	0.273
I think that health screening procedures are complicated.	1.115	0.700–1.788	0.647
Some health screening items leads to fear, anxiety, embarrassment, pain, and/or discomfort.	0.861	0.593–1.251	0.433
Some healthcare providers have negative attitudes.	0.779	0.499–1.288	0.274

Statistically significant is based on *p*-value < 0.05.

**Table 7 ijerph-17-03495-t007:** Factors that influence willingness to participate in health screenings.

Motivating Factors	Negative Factors
1. Conveniently located screening sites	1. Personal health belief (feel quite healthy)
2. Incentives (such as free health services or benefits)	2. Procedures were too complicated to understand (especially for elderly people)
3. Reasonable price	3. Waited for a long time or lack of time
4. Healthcare providers factors’ attitude and professional diagnostic services and facilities.	4. Physical pain or negative emotions (such as fear, anxiety, embarrassment and pain)
	5. Having had a negative experience during a previous health checkup

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
