# Peer review of "Why People Do Not Attend Health Screenings: Factors That Influence Willingness to Participate in Health Screenings for Chronic Diseases"

_ijerph, 2020, doi:10.3390/ijerph17103495_

Round 1
Reviewer 1 Report
No comments and suggestions.
Author Response
Thanks very much for your valuable comments~
Reviewer 2 Report
Although the manuscript is improved by adding the multivariable analyses, I still have questions about removing patients with chronic diseases from the sample. The N of the study is 204 patients. The author describe that 'almost one-third of the study population had a chronic disease', so the new N should be around 140 patients for all analyses. In the revised manuscript, the N is still 204, despite the authors state that 'Those who were diagnosed with chronic diseases and under routinely clinical treatments were excluded from this study (at line 75-76; page 2)'. Either they forgot to change the N of the sample in the manuscript, or the patients with chronic diseases are still not removed from the sample.
Author Response
Dear Reviewer/Professor
First, I would express my sincere appreciation for your positive comments and valuable suggestions those are supported me to make my work productive and stimulating without you this research work would not have been perfect possible.
This research was under careful IRB review, therefore follow the regulation, we have to exclude people who were diagnosed with chronic diseases and under routinely clinical treatments.
The sentence "almost one-third of the study population had a chronic disease" is study population who *suspected to have chronic diseases* (which means they were not officially announced to have chronic diseases). I think that’s my fault to make assumptions boldly about suspected some of interviewees might have chronic diseases by BMI/ life environmental/the habit of smoking or drinking….
I have revised the paper and go through again, please kind to see the attachment which marked in red color at page 2 line 75-76 and page 9 line 261-262.
Thank you again and wish you have a nice day!

Reviewer 3 Report
It reads better now.
Author Response

(The authors gave the same response as above.)

Round 2
Reviewer 2 Report
I did not understand that the authors meant that a part of the population is *suspected to have chronic diseases*. In that case, the proposed changes in the manuscript are sufficient in my opinion.
---------------------------
Previous version review process:
Round 1
Reviewer 1 Report
This paper may contribute to health screenings for chronic diseases. However, I consider that this paper requires a revision based on the following comments.
1. "It is estimated that, by 2050, 21 million people in the United States will be living with a chronic disease [11-12]. Worldwide, by 2020, more than 1 billion people had reached old age [16, 29-30]." The references were not numbered in order of appearance in the text.
2. You did not perform hypothesis testing between rural group and urban group, so it is difficult to say "The two groups (i.e., rural and urban) were very similar… ", "more rural group participants had undergone…"
3. The cut-off for underweight, standard, overweight, and obese should be described in the methodology section.
4. In line 67 on page 2 and in line 101 on page 3, there are two "Table 1". In line 68, this table has not an explanatory title.
5. In Pearson correlation analysis, two variables are required to be continuous variables. However, the variables in Table 4 are categorical variables.
6. You did not talk about limitations of your study. Information was based on self-report and was subject to recall and response bias. Selection bias may occur when selecting the study population.
Author Response
Point 1:
"It is estimated that, by 2050, 21 million people in the United States will be living with a chronic disease [11-12]. Worldwide, by 2020, more than 1 billion people had reached old age [16, 29-30]." The references were not numbered in order of appearance in the text.
Response 1:
- Thanks for your valuable suggestions. I have modified the manuscript according to your comments.
- By 2020, more than 1 billion people around the world had reached old age [1,2,3]. As a result of these changing demographics, since 2020, more than half of global healthcare expenditures have been related to the treatment of chronic conditions [1,4-10].
- Please find the revision (at line 37~; page 1).
Point 2:
You did not perform hypothesis testing between rural group and urban group, so it is difficult to say "The two groups (i.e., rural and urban) were very similar… ", "more rural group participants had undergone…"
Response 2:
- Thank you for your valuable suggestions.
- In addition, more rural group participants had undergone a previous health examination (74.5%) than had urban group participants (52.5%).
- Please find the revision (at line 123; page 3)
Point 3:
The cut-off for underweight, standard, overweight, and obese should be described in the methodology section.
Response 3:
- Thanks for your valuable suggestions.
- Participant weights were classified according to the body mass index (BMI), in which weight ranges include underweight (under 18.5 kg/m2), normal weight (18.5 to 25 kg/m2), overweight (25 to 30 kg/m2), and obese (over 30 kg/m2).
- Please find the revised manuscript for your reference. (at line 84; page 2)
Point 4:
In line 67 on page 2 and in line 101 on page 3, there are two "Table 1". In line 68, this table has not an explanatory title.
Response 4:
- Thanks for your valuable suggestions.
- Table 1 Clinical characteristics of urban and rural areas considered in this study (n=204).
- Please find the revision (at line 77; page 2).
Point 5:
In Pearson correlation analysis, two variables are required to be continuous variables. However, the variables in Table 4 are categorical variables.
Response 5:
- Thanks for your valuable suggestions.
- Please find the revision in Table 4. (at line 181~; page 6)
Point 6:
You did not talk about limitations of your study. Information was based on self-report and was subject to recall and response bias. Selection bias may occur when selecting the study
Response 6:
- Thanks for your valuable suggestions.
- Our findings are subject to three limitations. First, in this study, health-related behaviors were self-reported and therefore may be reporting bias. Second, findings of this study may not generalizable because we employed a simple random sample of 204 participants. Third, because rural areas have lower population densities than urban areas, we included subjects from three different rural areas.
- Please find the revision (at line 252~ page 8 )
Reviewer 2 Report
This is an interesting study about willingness to paticipate in health screening. This is an import subject in the light of the increasing number of patients with chronic diseases.
Major comments:
- It is unclear what the authors mean by the term 'health screening', since, in my opinion, the results of this study can differ between different chronic diseases. For instance, in cardiometabolic diseases weight and smoking are important risk factors, but for other diseases, like auto-immune diseases the risk factors are less clear and this could affect the willingness to participate.
- It would be better to exclude the patients who already have a chronic disease, because this will bias the results.
- Multivariable analyses are missing. This could provide information about the most important predictors of willingness to participate (for example, see the work of colleague Petter (doi: 10.1186/s12889-015-1379-0).
- The response rate is missing and it would be interesting to know whether the respondents were representative for the target population.
Minor comments:
- The introduction starts with numbers from the US, while the study is performed in Taiwan. This does not make sense.
- Line 49: "Health screening is considered the best way to reduce deaths and disabilities associated with chronic diseases". I do not agree this quote of the authors. Without proper treatment, screening is not effective. Please rephrase the sentence.
- Line 50: "However, many people do not participate in health screenings [34]; thus, many chronic diseases are not diagnosed before symptoms appear". What do you mean with this sentence? It is not possible to diagnose a patients when there no symptoms, unless there is a clear asymptomatic phase, which is not the case in most of the chronic diseases. Do you mean treatment of risk factors?
Response to Reviewer 2 Comments
Point 1:
It is unclear what the authors mean by the term 'health screening', since, in my opinion, the results of this study can differ between different chronic diseases. For instance, in cardiometabolic diseases weight and smoking are important risk factors, but for other diseases, like auto-immune diseases the risk factors are less clear and this could affect the willingness to participate.
Response 1:
- Thanks for your valuable suggestions. The term of health screening in this study, we focus on specific disease, as you can see in the table 1. Distribution of participants’ clinical characteristics. Based on these four areas’ background and occupational related factors, the residents at higher Risk for chronic Illness such as metabolic, lung, liver, and kidney diseases and various cancers. That’s the main reason why we pick these to be our research targets.
Point 2:
It would be better to exclude the patients who already have a chronic disease, because this will bias the results.
Response 2:
- Thanks for your valuable suggestions. In this study we proceed with simple random sampling for the target population. We aimed to identify the main reasons why people did not participate the community-based health screening service that have been provided by the Chang Gung Memorial Hospital for five consecutive years.
Point 3:
Multivariable analyses are missing. This could provide information about the most important predictors of willingness to participate (for example, see the work of colleague Petter (doi: 10.1186/s12889-015-1379-0).
Response 3:
- Thanks for your valuable suggestions. I have already modified according to your indication. Please find the revision in Table 4 (at line 181; page 6)
Point 4:
The response rate is missing and it would be interesting to know whether the respondents were representative for the target population.
Response 4:
- Thanks for your valuable suggestions. In this study we use a simple random sample, selected a sample of 204 participants for completing an one on one interview questionnaire. Please find the attachment of the revised version for your reference.
Point 5:
The introduction starts with numbers from the US, while the study is performed in Taiwan. This does not make sense.
Response 5:
- Thanks for your valuable suggestions.
- By 2020, more than 1 billion people around the world had reached old age [1,2,3]. As a result of these changing demographics, since 2020, more than half of global healthcare expenditures have been related to the treatment of chronic conditions [1,4-10]
- Please find the revision(at line 37; page 1)
Point 6:
Line 49: "Health screening is considered the best way to reduce deaths and disabilities associated with chronic diseases". I do not agree this quote of the authors. Without proper treatment, screening is not effective. Please rephrase the sentence.
Response 6:
- Thanks for your valuable suggestions.
- Routine health screening is considered to be one of the best ways to reduce deaths and disabilities associated with chronic diseases [31-33]
- Please find the revised version (at line 49~; page2)
Point 7:
Line 50: "However, many people do not participate in health screenings [34]; thus, many chronic diseases are not diagnosed before symptoms appear". What do you mean with this sentence? It is not possible to diagnose a patients when there no symptoms, unless there is a clear asymptomatic phase, which is not the case in most of the chronic diseases. Do you mean treatment of risk factors?
Response 7:
- Thanks for your valuable suggestions. Please find the revision (at line 50; page2)
- Health screenings can prevent and detect diseases in earlier, more treatable stages. This significantly reduces the risks posed by diseases, including disability and early death [34]
- We are trying to describe the value of a screening can be an effective way of reducing morbidity and mortality from disease by early detection. For example some studies have shown that screening average-risk individuals using Faecal Occult Blood Test (FOBT) can reduce colorectal cancer mortality by 15-20%.
Reviewer 3 Report
The study “Why people do not attend health screenings: Factors which influence willingness to participate in health screenings for chronic diseases” is a very important public health topic. The study has four strengths.
- The sample size is sufficient, and it allows for subgroup analyses and it is a plus to this study.
- The findings of the study are important contributions to public health
- The authors provided sufficient background information.
Major concern
- The study’s statistical analyses were very weak.
- The authors stated “… stepwise multiple logistic regression models were used to identify.” However, there is nowhere the authors reported the results of the logistic regression analyses. No odds ratio, no confidence interval except frequencies and percentages. I believe the study will be stronger if the multiple logistic regression analysis could be used to evaluate the differences between rural groups and urban groups with regards to their health screening behavior. It looks to me that the study was initially set up to investigate that, but the authors failed to execute or report that results. Is it because they were not significant?
- I am not sure why the authors decided to analyze or report individual items instead of aggregating those individual items to find a composite score for subscales. For example, the attitudes, beliefs of health status and health behavior factors could be grouped into individual subscales and used the composite score for each to do the analysis. Or the 10 items could be combined as one variable for the analysis. As it is now, it is fair to say that the authors just looked at the individual items and combined the agree and strongly agree responses and concluded that X number of participants report (eg. participants (47.5%; n=97) said that procedures involved in health screenings are too complicated and difficult to understand)
- The authors stated “… 32.8% of participants (n=67) reported that certain health screening procedures, such as mammograms or cervical smears.” However, in table 2 there is nowhere mammograms or cervical smears were mentioned. In other words, did not questionnaire specifically ask about mammograms or cervical smears screening procedure or just screening procedure in general? The authors need to be clear about this.
- My major concern is with table 4. The authors stated “The effects that individual factors were found to have on health behaviors are shown in table 4. Health behaviors were positively related to health attitudes and health awareness, which included health beliefs, anxiety, and self-reported health status (P≤0.01).” However, in table 4 there is no mention of anxiety, beliefs and health status. This categorization goes back to support my earlier point of combining individual items. The authors could have three main subscales namely anxiety, beliefs and health status variables. Since they did not combine them, I wonder which items they referred to as anxiety, beliefs and health status.
Minor concern
- The authors stated “Today, approximately 45% of all Americans (133 million) suffer from at least one chronic disease [1-5], and one in three adults has two or more chronic conditions [6-10]. It is estimated that, by 2050, 21 million people in the United States will be living with a chronic disease [11-12].” My question is Why concentrating on data from the US if the study is about Taiwan? The first two sentences suggest that the study is being done in the US. I completely understand you can make inferences if data are not available for your specific target population but it the case of this one, it is misleading. It is ok to use the statements but I suggest find an open sentence that will focus on your study population and use those two sentences to support your point.
- The authors stated, “This is particularly true in Taiwan, were an aging population has presented many challenges that have strained healthcare resources.” It should have been “This is particularly true in Taiwan, where …”
- The authors stated, “Sociodemographic and clinical characteristics of participants (by district) are shown in Table 1.” I am confused, under the Study design, sample, and recruitment section you have a table which is not clear what kind of table it is. Is it the table 1 you are referring to or you are referring to the table 1 in the results section? If you are referring to Table 1 in the results section, then the table in the study design section needs legend and it is confusing most especially the numbers at the end. Not sure which group Anle and Wanli alone or they are for more than one specific placer
- The recruitment method needs some clarifications or needs details. The mode of data collection is missing, hardcopy, electronically delivered, or how was the data collected?
- The authors stated “For this, participants were asked to indicate their level of agreement which a given statement using a 5-point Likert scale (whereby 1 represented “strongly disagree” and 5 represented “strongly agree”)” It should read “For this, participants were asked to indicate their level of agreement with …”
- The authors stated, “None of the participants had ever participated in a community-based screening for any chronic disease at the Chang Gung Memorial Hospital, which is one of the top-rated medical centers in Taiwan .” My question is Does it mean that they have had a screening for chronic diseases from another place? Why does screening from this hospital alone matter? If they received screening from another place, were they excluded from the study or include? I guess this sentence presupposes that any screening other than one received at this hospital doesn’t matter.
- The authors stated, “Furthermore, more than 50% of participants (58.33%; n=119) said that they had never heard of these community-based screening services.” Just like my concern above, Is this study assessing the community-based screening at one particular place or community-based screening in general? If the answer is yes, then this needs to be stated clearly in your purpose of the study. You set up the introduction and the purpose as a general screening study. If the study is about a specific screening you also need to talk a little bit about the program in your introduction section.
- The authors stated, “3.2.1. Correlation of health consciousness and health behaviors.” However, the results reported under this subheading do not establish any correlation. This heading is misleading. The authors just reported the proportion of the participants' perceptions and not correlation
Author Response
Point 1:
The authors stated “… stepwise multiple logistic regression models were used to identify.” However, there is nowhere the authors reported the results of the logistic regression analyses. No odds ratio, no confidence interval except frequencies and percentages. I believe the study will be stronger if the multiple logistic regression analysis could be used to evaluate the differences between rural groups and urban groups with regards to their health screening behavior. It looks to me that the study was initially set up to investigate that, but the authors failed to execute or report that results. Is it because they were not significant?
Response 1:
- Thanks for your valuable suggestions. We have modified the manuscript according to your indication (at line 102; page 3)
Point 2:
I am not sure why the authors decided to analyze or report individual items instead of aggregating those individual items to find a composite score for subscales. For example, the attitudes, beliefs of health status and health behavior factors could be grouped into individual subscales and used the composite score for each to do the analysis. Or the 10 items could be combined as one variable for the analysis. As it is now, it is fair to say that the authors just looked at the individual items and combined the agree and strongly agree responses and concluded that X number of participants report (eg. participants (47.5%; n=97) said that procedures involved in health screenings are too complicated and difficult to understand)
Response 2:
- Thanks for your valuable suggestions. In this regard, we tried to understand which factor influencing the decision making of non-participating in community-based health screening more. Thus, we can change the service model accordingly.
Point3:
The authors stated “… 32.8% of participants (n=67) reported that certain health screening procedures, such as mammograms or cervical smears.” However, in table 2 there is nowhere mammograms or cervical smears were mentioned. In other words, did not questionnaire specifically ask about mammograms or cervical smears screening procedure or just screening procedure in general? The authors need to be clear about this.
Response 3:
- Thanks for your valuable suggestions. Please find the revision of manuscript (at line 161; page 5).
Point 4:
My major concern is with table 4. The authors stated “The effects that individual factors were found to have on health behaviors are shown in table 4. Health behaviors were positively related to health attitudes and health awareness, which included health beliefs, anxiety, and self-reported health status (P≤0.01).” However, in table 4 there is no mention of anxiety, beliefs and health status. This categorization goes back to support my earlier point of combining individual items. The authors could have three main subscales namely anxiety, beliefs and health status variables. Since they did not combine them, I wonder which items they referred to as anxiety, beliefs and health status.
Response 4:
- Thanks for your valuable suggestions. Please find the revision of manuscript (at line 181; page 6)
Point 5:
The authors stated “Today, approximately 45% of all Americans (133 million) suffer from at least one chronic disease [1-5], and one in three adults has two or more chronic conditions [6-10]. It is estimated that, by 2050, 21 million people in the United States will be living with a chronic disease [11-12].” My question is Why concentrating on data from the US if the study is about Taiwan? The first two sentences suggest that the study is being done in the US. I completely understand you can make inferences if data are not available for your specific target population but it the case of this one, it is misleading. It is ok to use the statements but I suggest find an open sentence that will focus on your study population and use those two sentences to support your point.
Response 5:
- Thanks for your valuable suggestions.
By 2020, more than 1 billion people around the world had reached old age [1,2,3]. As a result of these changing demographics, since 2020, more than half of global healthcare expenditures have been related to the treatment of chronic conditions [1,4-10]
Please find the revision (at line 37; page 1)
Point 6:
The authors stated, “This is particularly true in Taiwan, were an aging population has presented many challenges that have strained healthcare resources.” It should have been “This is particularly true in Taiwan, where …”
Response 6:
- Thanks for your valuable suggestions. The first paragraph of the introduction has been rewritten (at line 37; page 1).
Point7:
The authors stated, “Sociodemographic and clinical characteristics of participants (by district) are shown in Table 1.” I am confused, under the Study design, sample, and recruitment section you have a table which is not clear what kind of table it is. Is it the table 1 you are referring to or you are referring to the table 1 in the results section? If you are referring to Table 1 in the results section, then the table in the study design section needs legend and it is confusing most especially the numbers at the end. Not sure which group Anle and Wanli alone or they are for more than one specific placer
Response 7:
- Thanks for your valuable suggestions. Please find the revision of manuscript, legends of Table1 (at line 77; page 2)
Point 8:
The recruitment method needs some clarifications or needs details. The mode of data collection is missing, hardcopy, electronically delivered, or how was the data collected?
Response 8:
- Thanks for your valuable suggestions. Please find the revision of manuscript in 2.1. Study design, sample, and recruitment (at line 67~; page 2)
Point 9:
The authors stated “For this, participants were asked to indicate their level of agreement which a given statement using a 5-point Likert scale (whereby 1 represented “strongly disagree” and 5 represented “strongly agree”)” It should read “For this, participants were asked to indicate their level of agreement with …”
Response9:
- Thanks for your valuable suggestions. Please find the revision of manuscript (at line 91; page 3)
Point 10:
The authors stated, “None of the participants had ever participated in a community-based screening for any chronic disease at the Chang Gung Memorial Hospital, which is one of the top-rated medical centers in Taiwan .” My question is Does it mean that they have had a screening for chronic diseases from another place? Why does screening from this hospital alone matter? If they received screening from another place, were they excluded from the study or include? I guess this sentence presupposes that any screening other than one received at this hospital doesn’t matter.
Response 10:
- In this study, we are interested to know why people did not attend the health screening hosted by Chang Gung Memorial Hospital even though they have launched for five consecutive years. We aim to evaluate the most common reasons for non-participation. Through this information, we can improve the participation rates of health screening in the future.
Point 11:
The authors stated, “Furthermore, more than 50% of participants (58.33%; n=119) said that they had never heard of these community-based screening services.” Just like my concern above, is this study assessing the community-based screening at one particular place or community-based screening in general? If the answer is yes, then this needs to be stated clearly in your purpose of the study. You set up the introduction and the purpose as a general screening study. If the study is about a specific screening you also need to talk a little bit about the program in your introduction section.
Response 11:
- Thanks for your valuable suggestions. Please find the revision of manuscript (at line 63; page 1).
Point 12:
The authors stated, “3.2.1. Correlation of health consciousness and health behaviors.” However, the results reported under this subheading do not establish any correlation. This heading is misleading. The authors just reported the proportion of the participants' perceptions and not correlation
Response 12:
- Thanks for your valuable suggestions. Please find the revision of manuscript (at line 128; page 4)
Round 2
Reviewer 2 Report
The revised manuscript is improved, however, I am not completely satisfied. There are still three issues left that I would like to address.
Minor comment:
- As described in my previous comments, I still disagree with the authors that screening results in a reduction of deaths and disabilities. After screening, appropriate preventive treatment is necessary. This should be added to the introduction section
Major comments:
- As the rationale of the study is to prevent the development of chronic diseases with health screening, it does not make sense to include patients who already have a chronic disease. These patients should be removed from the analyses, since they could bias the results of the study.
- See also my previous response, point 3: the authors added table 4 to the revised manuscript, including associations. This is not the same as a multivariable analysis. In a multivariable analysis, willingness to participate could be associated with all independent variables, including strength of the association with odds ratios. With these models it is also possible to define the most important predictors of willingness to participate (for more details: see the paper of colleague Petter).
Point 1:
As described in my previous comments, I still disagree with the authors that screening results in a reduction of deaths and disabilities. After screening, appropriate preventive treatment is necessary. This should be added to the introduction section.
Response 1:
- Thanks for your valuable suggestions.
- Routine health screening is considered to be one of the keys to reduce healthcare burdens associated with chronic diseases [31-33]. Health screenings can prevent and detect diseases in earlier, more treatable stages. After screening, appropriate preventive treatment is necessary. Please find the revision (at line 49-51; page 2)
Point 2:
As the rationale of the study is to prevent the development of chronic diseases with health screening, it does not make sense to include patients who already have a chronic disease. These patients should be removed from the analyses, since they could bias the results of the study.
Response 2:
- Thanks for your valuable suggestions.
- In Table 3, either in rural or urban area one third of the people interviewed were suspected to have chronic diseases even though they did not have the specific knowledge, this may affect their willingness. Please find the revision (at line 248-250; page 8)
- (limitations of the study) we employed a simple random sample of 204 participants, and one-third of the study population were found may have chronic diseases, which may cause biasto their willingness. Please find the revision (at line 256-258; page 8)
Point 3:
See also my previous response, point 3: the authors added table 4 to the revised manuscript, including associations. This is not the same as a multivariable analysis. In a multivariable analysis, willingness to participate could be associated with all independent variables, including strength of the association with odds ratios. With these models it is also possible to define the most important predictors of willingness to participate (for more details: see the paper of colleague Petter).
Response 3:
- Thanks for your valuable suggestions.
- In this study, a total of 204 data were collected randomly and participants were asked to complete a one-on-one interview and a questionnaire. Data currently were in terms of categorical variables, thus we applied descriptive statistical tests and Pearson correlation coefficients to identify variables which significantly influenced willingness to participate in health screenings. Data size in each category is still too small to conduct multivariable analysis, we will continue to improve the study and present data using multivariable analysis in the future.
Reviewer 3 Report
I appreciate that the authors taking the time to address the comments. If the authors could read over the manuscript and make minor corrections, especially typos that will be great.
Examples of typos include
- On line 102 the authors stated, "among subjects with higher education levels, f. Data were analyzed using descriptive statistical tests.." Not sure why f is in that sentence.
- On line 164, the authors stated, "and/or discomfort in patients In addition, a very low percentage of participants" There should be a period (.) between patients and In
Author Response
Point 1:
I appreciate that the authors taking the time to address the comments. If the authors could read over the manuscript and make minor corrections, especially typos that will be great.
Examples of typos include
- On line 102 the authors stated, "among subjects with higher education levels, f. Data were analyzed using descriptive statistical tests.." Not sure why f is in that sentence.
- On line 164, the authors stated, "and/or discomfort in patients In addition, a very low percentage of participants" There should be a period (.) between patients and In
Response 1:
- Thanks for your valuable suggestions.
- Please find the revision, at line 104; page 3 and line 166; page 5